# STASIS: REINFORCEMENT LEARNING SIMULATORS FOR HUMAN-CENTRIC REAL-WORLD ENVIRONMENTS

**Georgios Efstathiadis, Patrick Emedom-Nnamdi, Jukka-Pekka Onnela, Junwei Lu**
Department of Biostatistics, Harvard T.H. Chan School of Public Health
Boston, MA 02115, USA
`{gefstath,patrickemedom,onnela,junweilu}@hsph.harvard.edu`

**Arinbjörn Kolbeinsson**
Evidation Health
London, UK
`arinbjorn@evidation.com`

## ABSTRACT

We present on-going work toward building *Stasis*, a suite of reinforcement learning (RL) environments that aim to maintain realism for human-centric agents operating in real-world settings. Through representation learning and alignment with real-world offline data, Stasis allows for the evaluation of RL algorithms in offline environments with adjustable characteristics, such as observability, heterogeneity and levels of missing data. We aim to introduce environments the encourage training RL agents that are capable of maintaining a level of performance and robustness comparable to agents trained in real-world online environments, while avoiding the high cost and risks associated with making mistakes during online training. We provide examples of two environments that will be part of Stasis and discuss its implications for the deployment of RL-based systems in sensitive and high-risk areas of application.

## 1 INTRODUCTION

Reinforcement Learning (RL) is becoming increasingly popular for a variety of tasks, ranging from robotic control and autonomous driving to artificial intelligence in the gaming domain. Despite its potential, the lack of realistic simulators for RL agents operating in the real-world is a major limitation for the development of reliable agents. Current simulators lack the capability to model real-world applications of RL. This includes missing key components such as accounting for heterogeneity within the environment (specifically within the reward function) and observability, as all real-world environment are inherently perceived as partially observable. Furthermore, these simulators often lack the ability to handle missing data, either irregularly sampled data or observations missing at random (due design of the data collection tool used or observed features) and missing not-at-random (due to outcomes). Lastly, they lack the ability to generate observed data, as simulators should be thought of as a generative model of the real-world, where we want to generate samples close to the observed data.

The lack of realistic simulators for RL agents hinders the development of agents that can be successfully deployed to real-world tasks. The high cost and risk associated with inaccurate predictions during online training makes it an important problem to address. To this end, we introduce Stasis[1], a suite of RL environments that aim to maintain realism for human-centric agents operating in real-world environments. Through representation learning and alignment with real-world offline data, Stasis allows RL systems to be trained in offline environments with tunable characteristics, such as observability, heterogeneity and levels of missing data. The resulting RL agents are capable of maintaining a level of performance and robustness that is comparable to agents trained in real-world online environments, while avoiding the high cost and risk associated with making mistakes during online training.

---

[1]We plan on releasing the code for Stasis and for the two environments later this year.

**Related Work.** The most similar work to the one we present here is the Gymnasium, formerly known as OpenAI gym as seen in Brockman et al. (2016), and the Safety Gym by Ray et al. (2019). Both of these are suites of environments where RL agents can be trained without requiring real-world deployment. However, they both place emphasis on robotics and control, with Safety Gym making use of MuJoCo (Todorov et al., 2012) with a focus on constrained RL. The Stasis library which we introduce here will focus on open problems related to RL in healthcare, including partial observability, heterogeneity, missing data and make use of labelled real-world data through offline RL (Levine et al., 2020).

## 2 UNDERLYING FRAMEWORK & CONSIDERATIONS

On-going efforts to build simulated environments for benchmarking conventional and emerging RL algorithms center on emulating the realism and practicality of real-world settings. Evaluating algorithms in this fashion affords practitioners the ability to rigorously examine the suitability of an algorithm before initial deployment in the real world. In this paper, we identify four *core* themes that are important for representing *human-centric* real-world settings. Specifically, settings where the decision-making policy directly interacts with a human, or provides actions for a human to execute within their own environment.

**Observability.** Observability determines the full-range of information from the environment available to the agent for decision-making. In real-world settings, environments are typically partially observable; the agent only has access to a limited view of the current state of the environment (Littman, 2009). This can make it difficult to learn an optimal policy, as the agent may be missing important information or have to rely on incomplete observations to determine its actions. Therefore, a well-designed observability mechanism that captures the relevant information is critical for learning a good policy. However, increasing observability can also lead to higher computational and memory requirements, making it important to strike a balance between having enough information to make informed decisions and keeping the complexity manageable.

**Heterogeneity.** In real-world settings, the reward signal may vary between agents operating within a single environment. As such, learning a single policy that aims to optimize the reward for all agents is often difficult, leading to sub-optimal performance for select agents (Chen et al., 2022; Jin et al., 2022). Generally, this can result in a situation where some agents learn different, unintended behaviors. In multi-agent systems, this can lead to a lack of coordination, potentially hampering the functioning of the overall system. In applications such as healthcare where data from heterogeneous subjects are often used to make decisions for single subject, failing to account for heterogeneity in the reward signal can lead to an alignment problem, severely impacting the relevancy of the learned policy. Mitigating these challenges may require algorithms to leverage techniques from areas of research such as multi-agent reinforcement learning, or to directly modify the reward functions to account for the heterogeneity between agents.

**Missing Data.** The effectiveness of a policy in reinforcement learning is closely tied to the quality and quantity of data used to train the model. If the agent encounters missing data, such as incomplete or unavailable state or reward information, it may be unable to accurately estimate the value of different actions, leading to suboptimal decisions (Awan et al., 2022; Lizotte et al., 2008; Shortreed et al., 2011). Missing data can happen due to irregular sampling, where data is missing at random, which can occur due to various factors such as technical failures or data collection constraints. Additionally, data may be missing not at random, such as when specific actions or states are more likely to be absent. In healthcare application, this can be due to phenomena such as self-selection bias, where participants in the study exercise control over whether or not they participate in the study or how much data they provide. Therefore, it is essential to consider the consequences of missing data and address it using techniques such as imputation, data-augmentation, or other advanced methods for handling missing data in reinforcement learning (Shortreed et al., 2011; Awan et al., 2022).

**Offline Data.** Previously collected experiential data from agents interacting within a given environment can be used to enhance the robustness and reliability of the simulated environment. We envision that offline data can be used to improve the following aspects of the simulated environment: (1) *state representation* – offline data can be used to provide more accurate state representations for

the agent, including information about the environment, objects, and other agents (Zang et al., 2022); (2) *model dynamics* – the interactions between objects and agents in the environment can be modeled more accurately using offline data, allowing for a more realistic representation of the environment's dynamics Kidambi et al. (2020). Lastly, in most real-world environments, decision-making policies are rarely trained from scratch, rather offline data is commonly used to learn policies that achieve an acceptable level of performance (Levine et al., 2020). As such, incorporating available offline data into simulated environments allows for a pre-training phase, where policies are first initialized using offline data before being deployed within the environment.

# 3 THE SIMULATOR

The structure of the simulator is similar to the structure of the Gymnasium API (Brockman et al., 2016). An environment is pulled from the library's collection and then any type of agent can be trained using the simulated environment. Each environment has the same method structure, in order for the users to be able to switch among environments and train on different scenarios with ease.

The difference to the Gymnasium API is that the environments also share parameters related to problems found in real world applications, in order to make the environments more realistic and thus the agents more robust to real world data. When initializing an environment, the complexity of the problem will be specified, but also some parameters that are important in a healthcare context (Awrahman et al., 2022) and which are problematic in the collection and curation of healthcare data (Pezoulas et al., 2019). The shared parameters, when it is possible for an environment, will be able to tune aspects such as the heterogeneity of the simulation (Angus & Chang, 2021), incorporate missing data or have partial observability and add stochasticity or noise to the simulation. This can look different for every environment, but the purpose of the parameters is shared.

## 3.1 HEALTHY TRAVELING SALESMAN

The first environment is a simulated weighted travelling salesman problem (Lu et al., 2020). Studies have shown that certain environmental exposures are associated with healthcare bio-markers, e.g. greenspace exposure is associated with lower levels of depression (Klein et al., 2022). The problem this environment represents is finding the optimal routes to maximize or minimize a certain exposure related to the health of an individual. The environment is initialized by providing a set of coordinates, each of which has to be visited once, and an exposure type (e.g. greenspace or bars). Then, the environment will collect information on the possible routes that can be taken to visit each coordinate using the OpenRouteService API (Neis & Zipf, 2008) and the exposures around the locations of interest using the Overpass API (Olbricht, 2015) which both use data collected from OpenStreetMap or OSM for short (OpenStreetMap contributors, 2017). In the current implementation, one of the coordinates to be provided is the starting location and the rest of the coordinates are the ones that are visited, with the agent ultimately returning to the starting point. The task is thus finding the optimized circle in a graph, with complexity of the problem being increased by simply adding more coordinates.

The reward for each action is a weighted average between the distance covered and the time spent at exposure at each route, which is also tunable at input depending on what the agent should focus on optimizing. The possible exposure information is limited only by the possible types of locations that are collected from OSM. In terms of the parameters mentioned before on making the simulations more realistic to collected data, possible concepts discussed include modifying heterogeneity by adding different constraints in the possible actions of different simulated users. Some users have trouble moving large distances or want to avoid certain trigger areas, which can be encompassed in the reward function. In terms of missing data and stochasticity we can tune the amount of information and noise we see in the possible routes. They can also be encompassed in a way that matches what we see from GPS collecting devices like smartphones and smartwatches, where missing data are not missing at random, but there are certain time-periods for which data are not being collected by the smart devices (Barnett & Onnela, 2018).

The following figures 1 and 2 are examples of rendering of the environment, where the green areas are greenspace locations (Novack et al., 2018) collected by the Overpass API, the blue arrows are the coordinates of the locations to be visited and the red home arrow indicates the coordinates of the

starting and ending location. This is the visualization after an episode has been run using a Deep-QN agent (Mnih et al., 2013) trained on maximizing greenspace exposure on a set of 8 coordinates in Boston, MA (Figure 1) and a set of 6 coordinates in Bronx, NY (Figure 2).

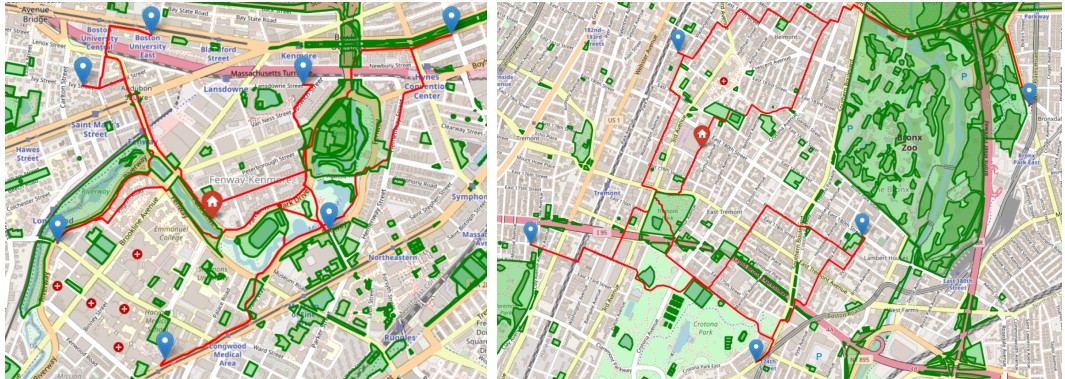

Figure 1: Map Environment rendering (Boston, MA)

Figure 2: Map Environment rendering (Bronx, NY)

Researchers that want to use this environment and possess offline GPS data can also encompass them to enrich the information in the environment and make it even more realistic (Gur et al., 2022). Using GPS trajectories, information can be collected on areas that people want to avoid or areas with more traffic and this information is reflected in the reward function of the environment. The GPS data can also be used to gain information of people's home and work locations or locations they like to visit frequently, thus making the environment adjust to a specific person's patterns and visit locations.

## 3.2 Resource Allocation in Clinical Settings

The second environment model in the Stasis library demonstrates a common problem encountered in clinical settings: dynamic resource allocation. This environment's properties can be highly complex due to the sophistication of modern clinical settings. In order to efficiently and operationally allocate resources, decision-making must be carried out on a case-by-case basis, considering the resources available, the individual conditions of multiple patients, and the associated costs and durations of the resources in question.

This environment's properties can be highly complex due to the sophistication of modern clinical settings. However, for its first iteration, it will be limited to a general setting. The state space includes the set of available resources and their characteristics, the occupancy of the clinical section, the time and date, other features that help forecast future occupancy, and relevant patient features, outcomes of utilized resources, and further diagnosis. The action space involves selecting resources from a given available set, which can be adjusted through domain expertise to incorporate best practices. The main goal of this framework is to understand the relationship between resources and patient outcomes and allow the agent to explore different strategies in the safe, non-destructive environment of the simulator.

## 4 Future Directions

As the initiative grows, it will be important to focus on community building. This can be accomplished by creating a leaderboard, hosting workshops, and adding existing standalone environments. Another focus of the library will be taking advantage of existing data to build more realistic environments. By leveraging existing offline data, the library could potentially use algorithms such as pretraining or initialization phases to further refine the environment and help it to behave in the most realistic way possible. Finally, there should be an active goal to make the environments relevant and useful in a medical or clinical context. To do this, researchers and developers will seek to collaborate with medical professionals to ensure the simulators are based on real world observations and are as accurate as possible. By doing so, Stasis can become a valuable tool for medical professionals.

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
