# OpenReview forum: "Stasis: Reinforcement Learning Simulators for Human-Centric Real-World Environments"
_ICLR.cc/2023/Workshop/TML4H — ICLR 2023 Workshop TML4H Poster_

### Official Review · Reviewer_WhGM · 2023-02-27
**Interesting identification of the limitations of current RL simulators. More details/figures on how STASIS addresses these limitations systematically would be helpful.**

**Rating:** 7
**Confidence:** 3

**Review:**

The paper proposes a new framework for simulating real-world environments for reinforcement learning (RL) agents. The authors argue that the current state-of-the-art RL simulators are not realistic enough to train agents that can be deployed in real-world applications. The authors identify limitations of current simulators and propose a new framework called Stasis that addresses these limitations. The authors provide examples of two environments that will be part of Stasis and discuss its implications for the deployment of RL-based systems in sensitive and high-risk sectors.
The paper is well-written and easy to follow.

#### Comments

- The paper does a good job in identifying the limitations of current RL simulators. However, the proposed framework remains not fully characterized.
   - Do the authors have a methodological/systematic way to identify and address the limitations of current simulators?
   - A figure summarizing the proposed framework and the main components would be extremely helpful.

- Do the authors plan to release the code base for STASIS and the environments?

---

### Official Review · Reviewer_CEyK · 2023-03-03
**It looks like an ongoing work and the paper lacks essential sections. I recommend to reject it in the current status.**

**Rating:** 3
**Confidence:** 4

**Review:**

The paper introduces Stasis, which can provide scenario-customized environments for reinforcement learning. Its goal is to simulate more realistic environments based on offline data and specific constraints of real-world tasks. Two examples are presented: the healthy traveling salesman problem and resource allocation in clinical settings. These examples demonstrate how Stasis can be customized to closely mimic the real-world environment. The paper also highlights the challenges encountered while implementing Stasis and provides intuitive solutions that are easy to understand and follow.

Weakness:
However, the paper did not present the practice of the proposed methods. Although the functions of Stasis sound promising, the paper lacks support from rigorous experiments. Besides this major weakness, the paper may improve its quality from several aspects:
1. The writing style needs to be more academic. Please consider using a third-role view instead of subjective sentences like "you will be able to xxx".
2. It's unclear how the first example, healthy traveling salesman, is related to the health topic. It seems like just a common traveling salesman problem. As the paper targets in health-domain, maybe more details about the challenge related to health need to be illustrated.
3. The scope of the paper is too large, thus making it lacks many necessary details about Stasis.
3. As the methods, experiments, and results sections are missing, it's hard to evaluate the effectiveness of this proposed suite. Please make up these necessary components.

---

### Meta-Review · Area_Chair_Fyg1 · 2023-03-05

**Recommendation:** Accept (Poster)
**Confidence:** 5

**Metareview:**

This paper develops a new framework called Stasis, which aims to provide more realistic environments for reinforcement learning agents. Overall, this paper is reasonably well-received, and both reviewers acknowledge the significance of the proposed framework. Some minor concerns are raised: the first reviewer suggests that the paper needs more details, and the second reviewer suggests that the paper needs more explanation of methodology, summary figures, and clarification of the code base.

The authors are highly encouraged to address the above (minor) concerns in the final version.